# Eggs Improve Plasma Biomarkers in Patients with Metabolic Syndrome Following a Plant-Based Diet—A Randomized Crossover Study

**DOI:** 10.3390/nu14102138

**Published:** 2022-05-20

**Authors:** Minu S. Thomas, Michael Puglisi, Olga Malysheva, Marie A. Caudill, Maria Sholola, Jessica L. Cooperstone, Maria Luz Fernandez

**Affiliations:** 1Department of Nutritional Sciences, University of Connecticut, Storrs, CT 06269, USA; minu.thomas@uconn.edu (M.S.T.); michael.puglisi@uconn.edu (M.P.); 2Department of Human Nutrition, Division of Nutritional Science, Cornell University, Ithaca, NY 14860, USA; ovm4@cornell.edu (O.M.); mac379@cornell.edu (M.A.C.); 3Department of Food Science and Technology, The Ohio State University, Columbus, OH 43210, USA; sholola.1@osu.edu (M.S.); cooperstone.1@osu.edu (J.L.C.); 4Department of Horticulture and Crop Science, The Ohio State University, Columbus, OH 43210, USA

**Keywords:** metabolic syndrome, eggs, plant-based diet, spinach, lipids, lutein, zeaxanthin, choline, TMAO

## Abstract

Plant-based (PB) diets are considered a healthy dietary pattern; however, eggs are not always included in this dietary regime. We hypothesized that the addition of two eggs per day would increase HDL cholesterol as well as plasma lutein, zeaxanthin and choline in individuals with metabolic syndrome (MetS). In this randomized controlled crossover intervention, we recruited 30 participants (49.3 ± 8 y) with MetS who followed a PB diet for 13 weeks. A registered dietitian advised all subjects on food selection and followed them through the intervention to ensure compliance. Participants underwent a 2-week washout with no eggs or spinach (a source of dietary lutein and zeaxanthin) and were randomly allocated to consume spinach (70 g) with either two eggs (EGG) or the equivalent amount of egg substitute (SUB) for breakfast for 4 weeks. After a 3-week washout, they were allocated the alternate breakfast. A total of 24 participants (13 women/11 men) finished the intervention. Plasma lipids, glucose, insulin, anthropometrics, plasma lutein, zeaxanthin, choline and trimethylamine oxide (TMAO) were assessed at baseline and the end of each intervention. When we compared individuals consuming the EGG versus the SUB breakfast, we observed a lower body weight (*p* < 0.02) and a higher HDL cholesterol (*p* < 0.025) after the EGG diet. There were no differences in plasma LDL cholesterol, triglycerides, glucose, insulin, or blood pressure. The number of large HDL particles measured by NMR was higher after EGG (*p* < 0.01) as compared to SUB. Plasma choline was higher in both treatments (*p* < 0.01) compared to baseline (8.3 ± 2.1 μmol/L). However, plasma choline values were higher in EGG (10.54 ± 2.8 μmol/L) compared to SUB (9.47 ± 2.7 μmol/L) *p* < 0.025. Both breakfasts increased plasma lutein compared to baseline (*p* < 0.01), while plasma zeaxanthin was only increased in the egg intervention (*p* < 0.01). These results indicate that consuming a plant-based diet in combination with whole eggs increases plasma HDL cholesterol, choline and zeaxanthin, important biomarkers in subjects with MetS.

## 1. Introduction

Current evidence from several studies demonstrates an association between consuming a plant-based (PB) diet and a reduced prevalence or risk of developing metabolic syndrome (MetS) [1,2,3]. However, vegetarian diets also document lower levels of high-density lipoprotein cholesterol (HDL-C) and higher levels of triglycerides (TG) [4]. The inclusion of eggs in a plant-based (PB) diet (lacto-ovo-vegetarian) may improve the metabolic conditions in MetS [5].

The modulatory effect of egg consumption on HDL-C composition in healthy [6] and MetS participants [7] is well-documented. Daily whole-egg consumption leads to more significant increases in plasma HDL-C and improvements in HDL profiles in MetS compared to intake of a yolk-free egg substitute during moderate carbohydrate restriction [8]. HDL being the primary transporter of lutein and zeaxanthin may impact the formation of HDL particles [9]. A positive association between plasma concentration of egg-yolk-derived carotenoids and lipoprotein particle size and HDL concentrations [9] suggests that eggs promote the formation of more large HDL particles that have been determined to be more functional [10]. Lifestyle interventions promoting low-fat, PB eating patterns have reduced HDL levels, with significant reductions in blood pressure, lipid profile and fasting blood glucose [11]. Based on this background, the addition of whole eggs to a PB diet will provide additional health benefits regarding plasma lipids.

MetS is associated with reductions in plasma lutein and zeaxanthin, and the altered composition of their lipoprotein transporters may affect disease risk [11,12,13,14]. Lutein and zeaxanthin protect cellular membranes and lipoproteins against oxidative-stress-induced reactive oxygen species (ROS) [15,16] and oxidized LDL. Lutein and zeaxanthin have been shown to reduce atherosclerosis in guinea pigs [17] and protect against age-related macular degeneration [18,19]. Previous observational and interventional studies suggest that lutein and zeaxanthin intakes are positively correlated with plasma HDL cholesterol and negatively correlated with plasma LDL cholesterol and triglyceride [20,21]. An intake of 6 mg/d of these carotenoids is sufficient to protect against oxidative stress, which leads to AMD or heart disease [22]. However, the dietary intake of lutein and zeaxanthin is very low in the American diet [23].

Raw spinach, one of the richest sources of lutein, contains about 20 mg of lutein plus zeaxanthin/100 g [24]. In contrast, most regular eggs typically contain about 0.3 mg of lutein + zeaxanthin/egg [24]. Even though the intake of green leafy vegetables increases serum lutein levels in healthy individuals, the bioavailability of this carotenoid is higher in eggs [25], possibly due to the lipid matrix of the egg yolk [12]. Given the ability of eggs to increase plasma lutein and zeaxanthin to a greater extent than leafy greens, eggs can be an important dietary source of carotenoids [26].

Eggs are an excellent source of choline in the form of phosphatidylcholine (PC), found in the egg yolk [27,28]. Choline is an essential nutrient for health because of its many functions in growth and development, neurological function and formation of membrane phospholipids, including phosphatidylcholine (PC) and sphingomyelin [29]. Betaine, derived from choline, has been shown to reduce fasting glucose in prediabetics [30], increase HDL in MetS [31], as well as reduce body fat and aid in weight loss [32]. Being a nutrient derived from either direct dietary intake or oxidation of choline, betaine is not considered an essential nutrient.

Intestinal microbes may metabolize dietary choline and betaine to form trimethylamine (TMA), which is subsequently absorbed and metabolized in the liver by the hepatic flavin-containing monooxygenase family of enzymes (FMOs) into trimethylamine N-oxide (TMAO) [33]. TMAO is recognized as a risk factor for cardiovascular disease (CVD) [33]. In contrast, betaine may decrease serum homocysteine concentration [34] associated with CVD risk and stroke [35]. However, these findings indicating the CVD risks associated with choline and its metabolites are inconsistent [36].

This study aimed to determine the role of whole eggs versus egg substitutes (described in methods) as part of a plant-based diet (excluding meat, poultry, fish and seafood) in the biomarkers that protect against oxidative stress and inflammation in patients with MetS. We hypothesized that the inclusion of whole eggs (EGG) in combination with spinach in a PB diet would increase plasma concentrations of lutein, zeaxanthin, and choline compared to an egg substitute (SUB) and spinach. Additionally, we hypothesized that the EGG intervention would increase plasma HDL and the formation of the larger HDL particles, which will allow for the transport of the increased carotenoids in plasma to the various tissues. While a plant-based diet is a promising dietary intervention to reduce the risks of MetS, the addition of whole eggs may elevate the benefits by providing a superior strategy to modulate biomarkers of disease.

## 2. Materials and Methods

### 2.1. Experimental Design

Thirty men and women aged 35–70 years and classified with MetS were recruited and enrolled in a randomized controlled crossover diet intervention. Recruitment started in August 2019 and was concluded in January 2022. The period encompassed more than 2 years due to the onset of COVID-12. We needed to stop the study for over 8 months before starting to recruit again. Participant recruitment was conducted through flyers and emails to the surrounding communities. All the subject interviews and the analysis for classification of MetS occurred in the Department of Nutritional Sciences at the University of Connecticut. Participants were included in the study if they met the National Cholesterol Education Program: Adult Treatment Panel (NCEP: ATP III) revised criteria for MetS [37] at screening and were willing to follow a PB diet for 13 weeks with either 2 eggs/d or the equivalent amount of egg substitute in combination with spinach as an omelet for breakfast. Exclusion criteria included liver disease, renal disease, diabetes, cancer, history of stroke, heart disease, glucose-lowering drugs or supplements and allergies to eggs or spinach.

All participants followed a lactovegetarian diet, restricting meat, poultry and fish. The PB diet was ad libitum, as there were no specific recommendations or restrictions for energy intake. A registered dietitian provided comprehensive dietary guidelines and instructions on following the PB diet through the intervention to ensure compliance. In addition to the PB diet, qualified participants underwent a 2-week washout period with no eggs or spinach (sources of dietary choline, lutein and zeaxanthin). They were randomly allocated to consume spinach (70 g) with either 2 eggs (EGG) or the equivalent amount of egg substitute (SUB) for breakfast for 4 weeks. After a 3-week washout, they were allocated the alternate breakfast. As a good source of lutein, zeaxanthin and choline, spinach was also provided with whole eggs or the egg substitute to evaluate whether the fats in the egg yolk would further aid in the absorption of these nutrients. Participants were asked to avoid eating eggs and spinach other than those provided by the study. All eggs (large, grade A, white) and egg substitutes (Egg Beaters original) were purchased at a local supermarket (Big Y, Tolland, CT) and provided to participants. The Egg Beaters original contained 99% egg whites and less than 1% of xanthan gum, guar gum and beta carotene for color and were fortified with vitamins. Frozen spinach (James Farm, Tolland CT) purchased from Restaurant Depot was portioned accurately and supplied each day. For further clarification, the content of dietary cholesterol, choline, lutein and zeaxanthin in EGG, SUB and spinach are presented in Table 1. Values are taken from USDA Nutrient Data [23].

We used information from our previous studies [6,9,25] to determine the sample size and calculated that 23 subjects would be sufficient to detect 10% changes in plasma lutein, zeaxanthin and choline with 80% power and an alpha of 0.05. The sample size was estimated using Graphpad StatMate2 software. We recruited 30 subjects to allow for attrition.

Participants were asked to maintain their regular physical activity, medications, and dietary-supplement usage upon starting the 13-week study. Compliance was monitored weekly through follow-up calls/texts by a registered dietitian and self-reported daily compliance forms. The intervention scheme is presented in Figure 1.

This study was approved by the Institutional Review Board, the University of Connecticut, Storrs under protocol H19-178 and registered at Clinicaltrials.gov (protocol NCT04234334). A total of 24 participants (*n* = 24; age = 49.3 ± 8 y) completed the 13-week study (13 women/11 men), and their data were used for the subsequent analyses. Informed consent was obtained from all participants before the screening.

### 2.2. Diet Analysis

The registered dietitian provided individualized dietary counseling, nutrition education materials, and sample menus and recipes. Participants filled out 3-day diet-intake and exercise records before and after the EGG or SUB intervention period (i.e., weeks 2, 6, 9 and 13 of the study period). Each 3-day record consisted of two nonconsecutive weekdays and one weekend day. Physical activity was expressed as the amount of time spent exercising in hours per day on concurrent days. Dietary records were analyzed using the Nutrition Data System for Research (NDSR) (Nutrition Coordinating Center, University of Minnesota, Minneapolis, MN, USA).

### 2.3. Blood Collection and Processing, 

Fasted blood was collected from participants after a 12 h overnight fast at 5 points, including screening, before and after each diet treatment (weeks 0, 2, 6, 9, 13). Antecubital venous blood samples were collected into EDTA-coated tubes and immediately centrifuged at 2000× *g* for 20 min at 4 °C for plasma separation. Blood samples collected in a serum tube with a clot activator and gel were left undisturbed for 20 min at room temperature to form a clot and centrifuged at 2000× *g* for 10 min at 4 °C for serum separation. Aliquoted plasma and serum samples were stored at −80 °C until analysis.

### 2.4. Anthropometrics and Blood Pressure (BP)

Height was measured on a stadiometer to the nearest 0.5 cm at screening. An electronic scale was used to measure weight, and it was recorded to the nearest 0.1 kg. BMI was calculated by dividing weight in kg by the square of height in meters (kg/m^2^). Weight, waist circumference (WC) and BP were measured at 5 points, including screening, before and after each diet treatment. WC was measured on bare skin during minimal respiration at the top of the iliac crest nearest 0.1 cm. BP was measured using an automated BP monitor (Omron, Healthcare Inc., Bannockburn, IL, USA) after participants were seated and rested for at least 5 min. The averages of three separate recordings were used for both WC and BP measures.

### 2.5. Plasma Lipids

Fasting plasma total cholesterol (TC), HDL-cholesterol (HDL-C) and triglycerides (TG),) were determined at baseline and the end of each breakfast period using an automated clinical chemistry analyzer (Cobas c 111, Roche Diagnostics, Indianapolis, IN, USA) via enzymatic and photometric detection methods as previously reported [7,21]. Plasma LDL-cholesterol (LDL-C) was estimated with the Friedewald equation [38].

### 2.6. Plasma Glucose, Insulin, HOMA-IR and MetS-Z Score

Plasma glucose was measured at baseline and after diet treatments using Cobas C111, an automated biochemical analyzer. Fasted plasma insulin was measured using a commercially available sandwich-enzyme-linked immunosorbent assay (ELISA) kit (Crystal Chem, IL, USA), which utilizes a specific antibody immobilized onto the microplate wells and an antibody labeled with Horseradish peroxidase (HRP) to detect and quantify insulin. The intra-assay variability was less than 5% for plasma insulin. The Homeostasis Model Assessment (HOMA-IR) equation estimated basal insulin resistance based on fasting plasma insulin and plasma glucose measurements [39].

MetS severity Z scores were calculated for participants at baseline and after EGG and SUB treatments using sex- and race/ethnicity-based formulas [40]. These scores were derived from a confirmatory factor analysis examining the correlation of the 5 criteria for MetS (waist circumference, systolic blood pressure, triglycerides, HDL cholesterol, fasting glucose) on nationally representative data from the National Health and Nutrition Examination Survey (NHANES) for adults 20–64 years of age. The weighted contribution of each of the MetS components to a latent MetS “factor” on a sex- and race/ethnicity-specific basis was determined (found at http://mets.health-outcomes-policy.ufl.edu/calculator/, accessed on 14 May 2022). 

### 2.7. Lipoprotein Particle Size and Subfractions

Plasma VLDL, IDL, LDL, and HDL particle concentrations and average particle diameters were quantified using nuclear magnetic resonance (NMR) spectroscopy at baseline and end of both treatments (weeks 6 and 13). NMR relies upon the unique resonance signal broadcasted by each particle following exposure to a magnetic field to separate the particles by size [41]. This analysis separated VLDL, LDL and HDL into small, medium and large subclasses and provided information on total lipoprotein particle concentration and mean particle size.

### 2.8. Plasma Choline and TMAO

Plasma choline and its metabolites—betaine, dimethylglycine (DMG) and TMAO—were quantified by stable-isotope dilution liquid chromatography with tandem mass spectrometry (LC/MS/MS), as previously described [42,43]. Plasma collected at the baseline and end of each intervention was used for these analyses.

### 2.9. Plasma Lutein and Zeaxanthin

The analysis of carotenoids in plasma was conducted using an Agilent 1260 HPLC-DAD and chromatographed using a C30 column (4.6 × 250 mm, 3 μm, YMC Inc., Wilmington, DE, USA). A total of 500 μL of plasma was required to extract lutein and zeaxanthin [44,45]. Both carotenoids were quantified at 450 nm using authentic external standard curves.

### 2.10. Statistical Analysis

All variables were analyzed using SPSS for Windows Version 25 (IBM Corp). Repeated-measures ANOVA was used to evaluate differences between baseline and at the end of each dietary period, treatment in plasma lipids, glucose, insulin, insulin resistance, choline metabolites, plasma TMAO, lutein and zeaxanthin, with time being the repeated measure (baseline and end of each dietary treatment); the data were reported as mean ± SD. *p* < 0.05 was considered to be significant. Finally, Pearson correlations were calculated between plasma biomarkers and lipoproteins. Sex, age, race and baseline Met-z score were used as covariates for plasma choline and its metabolites.

## 3. Results

### 3.1. Baseline Characteristics of Participants

Thirty participants were recruited and six dropped out of the study due to the sudden onset and widespread impact of COVID-19. Participants (*n* = 24) who completed the study were middle-aged (49.3 ± 8 y) and in the overweight-to-obese BMI range, consisting of 21 Caucasians and 3 African Americans. Approximately half of the participants that completed the study were females. Characteristics of participants at baseline are presented in Table 2.

### 3.2. Dietary Intake

The dietary and exercise patterns remained relatively consistent throughout the intervention. All participants complied with the PB diet and consumed their EGG or SUB with spinach for breakfast. Diet intake at baseline and the end of each dietary period is presented in Table 3. There was no difference in kcalorie (kcal) intake throughout the intervention. The fat intake (% kcal) was higher during EGG, while protein intake (% kcal) was higher during EGG and SUB treatments compared to baseline. The carbohydrate intake (% kcal) and added sugar intake were higher at baseline compared to the EGG and SUB breakfast periods. Predictably, cholesterol was higher during the EGG period, along with SFA and MUFAs. Omega 3 fatty acids were increased in both EGG and SUB periods compared to baseline. Dietary choline was significantly increased after EGG, whereas betaine was increased in SUB and EGG. The dietary intake of βcarotene, lutein, zeaxanthin, vitamins A, B2, and selenium were higher during the EGG and SUB periods. There were no differences in fiber intake, glycemic index, or glycemic load among treatments. EGG period also showed higher vitamin D and B12 intakes.

### 3.3. BMI and Weight Anthropometrics, Blood Pressure, Plasma Lipids and Glucose

During the EGG treatment, there was a moderate but significant reduction in weight and BMI compared to SUB and baseline (Table 4). Waist circumference and systolic and diastolic BP were unchanged for the duration of the intervention. As expected from previous interventions, HDL-c increased after EGG intake. At the same time, TG, LDL-C and parameters of glucose metabolism such as plasma glucose, insulin and homeostasis model assessment (HOMA-IR) remained unchanged. The LDL/HDL ratio, a commonly used marker of atherosclerosis risk, did not significantly change among treatments. The MetS-Z score placed our subjects on the 75% percentile of severity. There were no significant differences between baseline or at the end of the dietary treatments. The lack of differences might be due to the fact that subjects were on the PB diet for 2 weeks (baseline values).

### 3.4. Lipoprotein Particle Size and Subfractions

As indicated in Table 5, there were no significant changes in the number of totals, large, medium and small VLDL particles; in the number of IDL and the number of totals, large or small LDL particles. In contrast, participants had increases in total HDL particles (*p* < 0.025) as well as large HDL particles (*p* < 0.025) (Table 5). There were also no changes in lipoprotein size among treatments (Table 5). The number of large HDL particles for baseline, EGG and SUB is shown in Figure 2.

### 3.5. Plasma Choline, Metabolites and TMAO

Plasma choline concentration was higher after EGG treatment than baseline and SUB (Table 6); the choline metabolites, betaine and methionine, remained unchanged, DMG was significantly increased after EGG and SUB compared to baseline. Plasma TMAO concentration was not different between baseline and EGG, while SUB was higher than baseline.

### 3.6. Plasma Lutein and Zeaxanthin

Plasma lutein concentration increased significantly after EGG (475.7 ± 220.3) as well as SUB (461.9 ± 188.6) compared to baseline (282.7 ± 112.9) (Figure 3). There were significant increases in plasma zeaxanthin concentration after EGG treatment (93.5 ± 50.8) compared to SUB 73.1 ± 38) and baseline (68.6 ± 34.6).

Changes in plasma zeaxanthin were positively associated with the large HDL particles during the EGG period (r = 0.57, *p* < 0.01) (Figure 4). This was a strong correlation since the r was almost equal to 0.6 in a group of 24 participants.

## 4. Discussion

This study demonstrated that combining whole eggs with the PB diet exerts a protective effect in MetS participants, documented by the lower BMI, weight and increases in plasma HDL, zeaxanthin and choline. In addition to the high-quality protein, the egg yolk present in the EGG treatment provided essential nutrients such as choline, lutein, zeaxanthin and vitamin A, D and B2, as well as B12 and fats that aid bioavailability, confirming that a lacto-ovo-vegetarian diet proves to be a well-rounded and nutritious approach when compared to a lactovegetarian diet in individuals with MetS.

### 4.1. Dietary Changes Associated with Egg Intake

The first meal of the day, breakfast, may regulate and determine the energy intake at a subsequent meal [46]. Eggs are packed with high-biological-value protein that improves satiety, reducing food intake later in the day [47]. Compared to cereals, egg breakfasts provide overall satiety, reduced postprandial glycemic response and subsequent food intake in lean, young adults [47]. Likewise, when compared to an oatmeal breakfast, two eggs per day for breakfast increase satiety throughout the day in a young, healthy population without adversely affecting the biomarkers associated with CVD risk [46]. In our study, the significant reduction in the daily intake of dietary carbohydrates during the EGG and SUB periods may be due to the satiety developed by the nutrient-dense egg omelets. The lactovegetarian diet might have primarily depended on a carbohydrate source such as a bagel and cream cheese, oatmeal or cereals during baseline, which explains the significant increases in added sugar at baseline. The SUB is yolk-free but has added beta carotene for natural color and is fortified with vitamin A, D, E, B2 and B12, which explains the corresponding increases in their dietary intake. Eggs being the richest and most bioavailable source of choline resulted in significant increases in dietary choline. Betaine, another critical nutrient, is abundant in spinach (600–645 mg/100 g serving) [48] resulting in significant increases after SUB and EGG periods and not at baseline.

### 4.2. Anthropometrics and Blood Pressure

The significant decreases in weight and BMI after EGG may be because of the satiety provided by the egg breakfast, which reduced the carbohydrate intake. Low-carbohydrate diets are associated with increased satiety [49] and have proved effective for weight loss and weight management [50,51]. Nutrient-rich eggs, as well as the PB diet, may have contributed to this change.

### 4.3. Lipoprotein Modifications and Egg Intake

Although some studies suggest that egg consumption may be linked to elevated levels of blood cholesterol or diabetes mellitus [52], recent epidemiological studies do not agree with these findings [53,54,55,56,57]. In addition to and consistent with our previous findings [58,59], we did not find significant changes in total cholesterol, LDL-C, LDL/HDL ratio, or triglycerides. Similarly, plasma glucose and insulin remained constant after two eggs per day. HDL-C was significantly increased after the EGG period, thereby reversing one criterion of MetS.

Through its ability to remove excess cholesterol from peripheral tissues and transport them to the liver for excretion from the body via the reverse cholesterol transport pathway (RCT), HDL improves CVD [60]. In addition to RCT, HDL promotes antiatherogenic effects through its anti-inflammatory and antioxidative functions by protecting against endothelial dysfunction [61], inhibition of LDL-induced monocyte transmigration [62], as well as inhibition of LDL oxidation [62]. It has been previously reported that apart from increasing plasma HDL, daily whole-egg consumption promotes favorable shifts in HDL lipid composition and function [7,63]. There were no differences in total lipoprotein number or concentrations of lipoprotein subclasses between the treatments or baseline, including the atherogenic lipoproteins large VLDL, IDL and small LDL. Thus, the significant increases in total HDL and large HDL after consuming two eggs/d with the PB diet showcases the atheroprotective effect. The large HDL produced by eggs has been shown to have a higher cholesterol-effluxing capacity, demonstrating that it is more effective in RCT [7,63].

### 4.4. Carotenoids, Choline and Egg Intake

Dietary interventions have documented that whole-egg consumption increases dietary intake and improves plasma carotenoids in overweight individuals as part of a carbohydrate-restricted diet [64,65], in young men and women [66], metabolic syndrome participants [8,48], pre-menopausal women [25], elderly adults with altered lipid profile [67,68], after intake of regular, lutein-enriched and n-3 fatty-acid-enriched eggs [69] as well as in a healthy lacto-ovo-vegetarian population predominately on a plant-based diet consuming both n-3 fatty acid-enriched eggs and organic eggs [69]. In most of the above studies, plasma choline increased after consumption of eggs, whereas the choline metabolites’ TMAO levels did not increase [7,67].

The increases in dietary and plasma lutein and zeaxanthin after EGG treatment is consistent with previous findings [9,70]. Because of their association with the lipid matrix of the egg yolk, these lipophilic carotenoids have optimal gastrointestinal uptake, making them highly bioavailable [70] compared to other food sources. Spinach, a rich source of lutein, was cooked with oil for the omelets. Compared to baseline, this added fat might have contributed to its absorption and increased plasma lutein after SUB intake. However, zeaxanthin was higher during the EGG period because spinach may not be a good source of this carotenoid. The large HDL particles generated by egg intake were associated with plasma zeaxanthin, demonstrating the role of HDL particles in the transport of this carotenoid.

The 2020–2025 Dietary Guidelines for Americans list choline as an under-consumed nutrient in the American diet [71]. A significant number of adults and children fail to meet the adequate intake (AI) for choline in the US [72], which may increase the risk of adverse effects, including cognitive impairment, neural tube defects, muscle damage and fatty liver [73]. Due to the impact of choline deficiency on health issues, adequate consumption of choline becomes very important. The most concentrated sources of choline are of animal origin (example; 3 oz beef liver, 356 mg; 1 large hard-boiled egg, 147 mg), which raises the risk for choline deficiency while following a PB diet. Consumption of two whole eggs helps meet at least half the excellent intake recommendation (550 mg/day of choline for men and 425 mg/day for women) [73].

We found a significant increase in DMG and betaine after EGG and SUB intake compared to baseline. Betaine concentration (mg/100 g) is highest in plant foods such as wheat bran, wheat germ, spinach., pretzels and wheat bread [74]. Low plasma betaine concentrations are associated with an unfavorable cardiovascular risk profile in the MetS population [75] and an increased risk of secondary heart failure and acute myocardial infarction [75]. The dietary betaine and choline-derived betaine might have significantly increased DMG after EGG and SUB intake. We did not observe higher TMAO concentrations compared to baseline in agreement with previous studies in our lab that have shown increases in plasma choline with no significant increases in TMAO after three eggs per day [7,48]. The higher concentrations of TMAO after the SUB breakfast may be due to the increased dietary betaine from spinach. Although those values are significantly different compared to the baseline, they are still low compared to other reported values of TMAO for individuals with MetS [76]. Slight changes in microbiota associated with the PB diet might be responsible for these increases.

### 4.5. Strengths and Limitations

This study is unique and the first of its kind to better understand the contribution of eggs to a healthy diet such as a plant-based diet. Although all participants were habitually omnivores, they adhered to the diet with full compliance. The service of the registered dietitian motivated the participants, which was our biggest strength. We also provided the intervention foods and ensured that subjects did not change their physical activity or medications (dose/type) during the 13-week intervention.

The onset of COVID-19 posed several challenges in the study, such as the enrollment of the participants from this at-risk population, dropouts and delays in completion. The small sample size who completed the intervention and short duration (4 weeks), which might not have been sufficient to determine changes in some of the measured parameters, would be the limitation of this study. Another perceived weakness is the lack of blinding of the participants. The participants had no calorie restriction, so additional studies with calorie and weight maintenance would further clarify the effects on other criteria of MetS.

The novelty of this dietary intervention is that it answers existing questions and concerns regarding the health impacts of egg intake in the MetS population. Some individuals who follow a vegetarian diet choose to avoid eggs without being aware of the many benefits that this food can provide in improving dyslipidemias, reducing oxidative stress, and reducing inflammation. This study demonstrates that consuming whole eggs in combination with a plant-based diet offers a healthier dietary pattern when compared to egg substitutes by favorably affecting plasma lipids and antioxidant carotenoids, as well as choline, thereby reducing disease risk.

## 5. Conclusions

Based on the results from this study, we conclude that the PB diet with whole eggs (Lacto-ovo-vegetarian) appears to either maintain and improve dyslipidemias and the markers of oxidative stress and inflammation over vegan and lactovegetarian diets in participants with MetS. Eggs had a two-fold benefit for MetS patients. First, they had an increase in HDL cholesterol, which has been demonstrated to be protective against cardiovascular disease risk, and the large HDL is associated with a more effective reverse cholesterol transport. In addition, eggs did not result in higher LDL-C, TG or glucose. Second, eggs also resulted in high concentrations of plasma lutein and higher concentrations of zeaxanthin and choline, which are low in this population [15]. These compounds may protect patients with MetS against chronic diseases, including cardiovascular diseases, AMD, type II diabetes and Alzheimer’s.

Future studies should include larger groups of participants with more diversity in race/ethnic groups, as well as participants with different socioeconomic characteristics, including level of education, purchasing power and adherence to healthy lifestyles.

## Figures and Tables

**Figure 1 nutrients-14-02138-f001:**
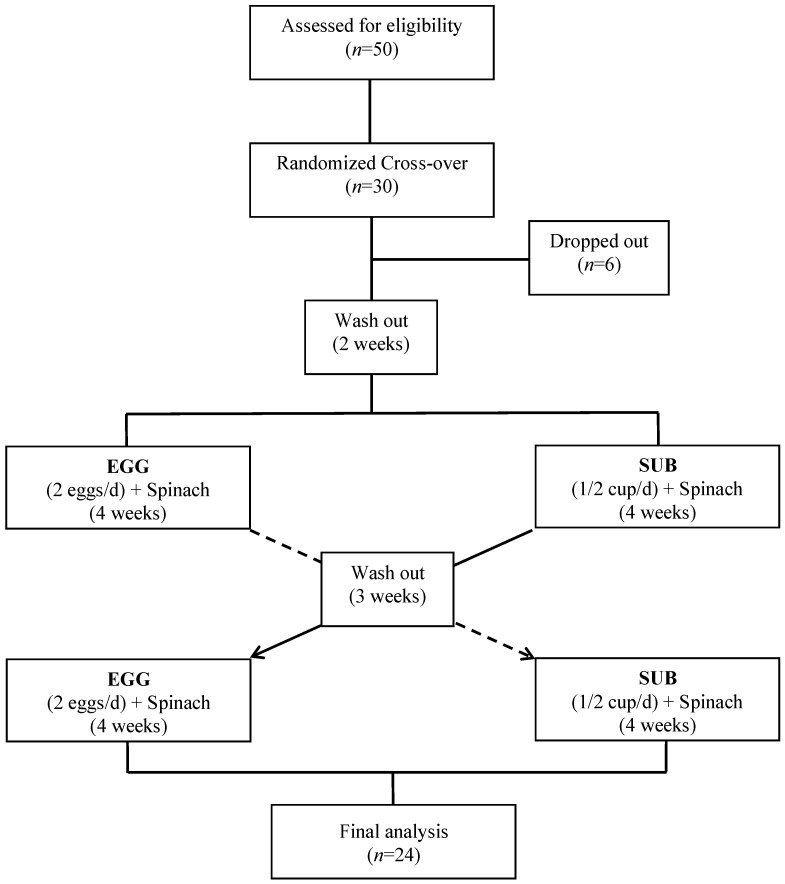
Experimental design. Participants underwent a 2-week washout period before being randomly allocated to either 2 eggs/d or equivalent egg substitute (1/2 c/d) for 4 weeks. Following a 3-week washout period, they were assigned the alternate treatment.

**Figure 2 nutrients-14-02138-f002:**
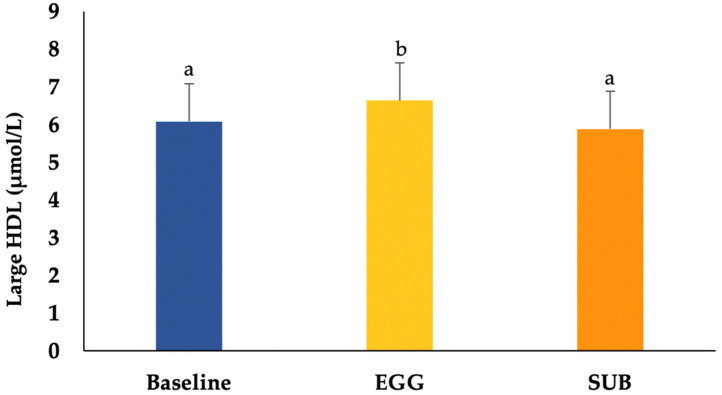
Concentrations of large HDL at baseline and after the intervention. Values are presented as mean ± SD. Values in the same row with different superscripts (a, b) are significantly different at a *p* < 0.01. The EGG breakfast resulted in higher concentrations of large HDL.

**Figure 3 nutrients-14-02138-f003:**
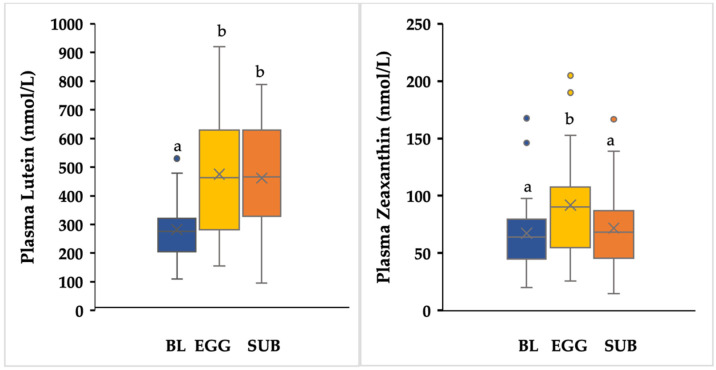
Concentrations of lutein and zeaxanthin at baseline (BL) and following the EGG and SUB intervention. Both EGG and SUB interventions resulted in higher concentrations of lutein (*p* < 0.01), but only EGG increased plasma zeaxanthin (*p* < 0.01) as indicated by different superscripts (a,b). Values are presented as mean ± SD.

**Figure 4 nutrients-14-02138-f004:**
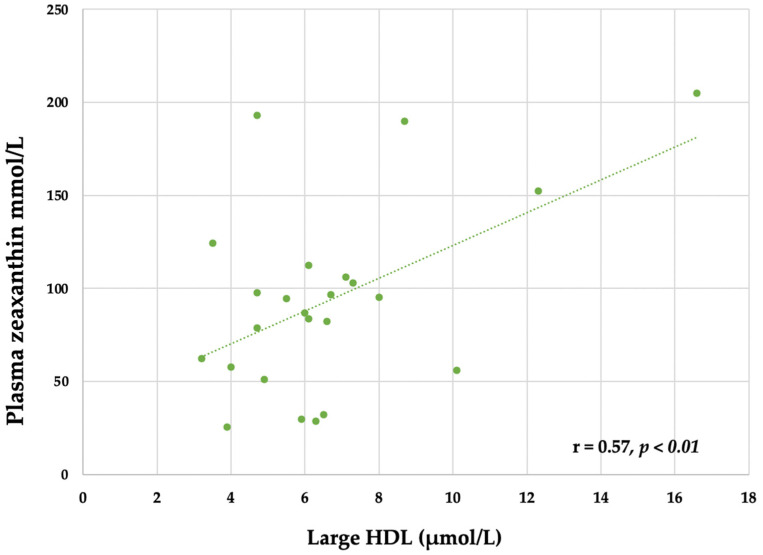
The relationship between large HDL and plasma zeaxanthin during the EGG period (Pearson’s correlation coefficient, r = 0.57, *p* < 0.01).

**Table 1 nutrients-14-02138-t001:** Daily intake of cholesterol, choline and lutein + zeaxanthin content in eggs, egg substitute (SUB) and spinach.

	EGG (2 Large Eggs)	SUB (½ cup Egg Substitute)	Spinach (70 gm)
Cholesterol (mg)	370	0	0
Choline (mg)	294–285	0	13.5
Lutein + Zeaxanthin (mg)	0.2–0.3	0	20.3

**Table 2 nutrients-14-02138-t002:** Baseline characteristics of participants ^1^.

Parameter	Values
Age (years)	49.3 ± 8
Gender (F/M)	13/11
Race (Caucasians/African Americans)	21/4
BMI (kg/m^2^)	34.3 ± 4.6
Gender Female (%)	54%
Waist Circumference (cm)	112.5 ± 11.9
Systolic Blood pressure (mmHg)	183 ± 27.6
Diastolic Blood pressure (mmHg)	86.6 ± 5.6
HDL cholesterol (mg/dL)	42.1 ± 10.3
Triglycerides (mg/dL)	155 ± 68
Glucose (mg/dL)	103 ± 12

^1^ Values are presented as mean ± SD.

**Table 3 nutrients-14-02138-t003:** Energy, macronutrients, fatty Acids, cholesterol, glycemic index, glycemic load, dietary fiber, carotenoids, vitamins, and physical activity at baseline and after the EGG or SUB breakfasts determined by dietary analysis using NDSR *.

Dietary Component	Baseline	EGG	SUB
Energy (Kcal) ^2^	1677 ± 573 ^a^	1798 ± 579 ^a^	1699 ± 512 ^a^
Total fat (%)	35.6 ± 6.5 ^a^	40.9 ± 7.2 ^b^	35.4 ± 6.4 ^a^
Total CHO (%)	49.6 ± 4.7 ^a^	43.7 ± 7.7 ^b^	47.1 ± 7.9 ^ab^
Total protein (%)	13.3 ± 3.0 ^a^	14.8 ± 2.9 ^b^	15.4 ± 2.8 ^b^
SFA (g)	21.8 ± 2.8 ^a^	29.2 ± 1.3 ^b^	24.2 ± 11.2 ^ab^
MUFA (g)	23.5 ± 9.3 ^a^	28.6 ± 10.9 ^b^	23.4 ± 9.1 ^a^
PUFA (g)	16.6 ± 5.8	18.2 ± 7.7	17.1 ± 7.5
TFA (g)	1.5 ± 0.8	1.9 ± 1.5	1.9 ± 1.3
Cholesterol (mg)	102 ± 86 ^a^	438 ± 135 ^b^	143 ± 115 ^a^
Omega-3 fatty acids (g)	1.6 ± 0.7 ^a^	2.0 ± 0.9 ^b^	2.0 ± 0.7 ^b^
Added sugars (g)	45.7 ± 36.2 ^b^	32.4 ± 30.0 ^a^	32.0 ± 21.2 ^a^
Glycemic Index	57.5 ± 4.3	56.5 ± 4.9	55.8 ± 4.8
Glycemic Load	112 ± 45	102 ± 42	101 ± 33
Fiber (g)	23.7 ± 8.7	21.6 ± 6.3	25.3 ± 8.6
Alpha-carotene	526 ± 489	452 ± 618	512 ± 594
Beta-carotene (µg)	4084 ± 2890 ^a^	6357 ± 2527 ^b^	7684 ± 3387 ^b^
Lutein + Zeaxanthin (µg)	3151 ± 4382 ^a^	9190 ± 1527 ^b^	9179 ± 2188 ^b^
Choline (mg)	200.7 ± 82.9 ^a^	436.0 ± 96.9 ^b^	226.7 ± 109.4 ^a^
Betaine (mg)	120.7 ± 65.7 ^a^	170.7 ± 65.6 ^b^	177.5 ± 82 ^b^
Vitamin A (µg)	1207 ± 538 ^a^	1643 ± 493 ^b^	1748 ± 640 ^b^
Vitamin D (µg)	3.3 ± 2.3 ^a^	5.4 ± 2.3 ^c^	4.3 ± 1.7 ^b^
Vitamin E (mg)	11.3 ± 6.3	11.8 ± 4	12.7 ± 3.9
Vitamin B2 (mg)	1.9 ± 0.8 ^a^	2.4 ± 0.7 ^b^	3.1 ± 0.5 ^c^
Vitamin B12 (mg)	2.9 ± 2.2 ^ab^	3.4 ± 1 ^b^	2.8 ± 1.4 ^a^
Sodium (mg)	2646.5 ± 1114.5	3189.5 ± 1099	3081 ± 994.2
Selenium (µg)	78.6 ± 38.9 ^a^	106.6 ± 29.4 ^b^	101.0 ± 29.3 ^b^
Physical activity (min)	53.2 ± 25.4	51.5 ± 21	47.8 ± 19.8

* Data are presented as mean ± SD, (n = 24). ^2^ Values in the same row with different superscripts (^a–c^) are significantly different at a *p*-value < 0.001. Abbreviations: CHO = carbohydrates; SFA = saturated fatty acids; MUFA = monounsaturated fatty acids; PUFA = polyunsaturated fatty acids; TFA = trans fatty acids.

**Table 4 nutrients-14-02138-t004:** Participant characteristics, anthropometrics, blood pressure (BP), waist circumference (WC) and plasma biomarkers of (*n* = 24) MetS participants at baseline and after EGG and SUB treatments for 4 weeks each *.

Parameters	Baseline	EGG	SUB
Body weight (Kg)	99.4 ± 19.6 ^b^	98.5 ± 19.2 ^a^	99.6 ± 20.1 ^b^
BMI (kg/m^2^)	34.3 ± 4.8 ^b^	33.8 ± 4.6 ^a^	34.7 ± 4.6 ^b^
Waist circumference (cm)	112.5 ± 11.9	113.4 ± 13.3	113.3 ± 12.7
Diastolic BP (mm Hg)	86.6 ± 5.6	86.2 ± 8.4	86.7 ± 6.6
Systolic BP (mm Hg)	183.0 ± 27.6	185.3 ± 29.0	179.1 ± 24.6
HDL cholesterol (mg/dL)	42.1 ± 10.3 ^b^	43.3 ± 10.7 ^a^	41.5 ± 10.1 ^b^
Triglycerides (mg/dL)	155 ± 68	149 ± 58	156 ± 66
LDL cholesterol (mg/dL)	109.9 ± 26.6	112.3 ± 25.9	108.1 ± 19.8
LDL/HDL ratio	2.75 ± 0.88	2.72 ± 0.77	2.72 ± 0.73
Glucose (mg/dL)	103 ± 12	93 ± 11	92 ± 9
Insulin (pmol/L)	67.68 ± 34	67.42 ± 34.66	71.3 ± 39.46
HOMA-IR	2.61 ± 1.41	2.62 ± 1.54	2.71 ± 1.62
MetS-Z score	0.75 ± 0.40	0.70 ± 0.49	0.74 ± 0.50

* Values are presented as mean ± SD. Values with different superscripts differ at *p* < 0.05 as s determined by repeated-measures ANOVA with LSD post hoc analysis.

**Table 5 nutrients-14-02138-t005:** Lipoprotein particle concentrations and subfraction analysis from fasting plasma of MetS participants (*n* = 24) at baseline and after EGG or SUB diet for 4 weeks each *.

Lipoprotein Concentration	Baseline	EGG	SUB
Total VLDL (nmol/L)	62.0 ± 17.8	68.7 ± 31.5	68.0 ± 25.4
Large VLDL (nmol/L)	8.7 ± 5.0	8.7 ± 5.7	10.9 ± 8.1
Medium VLDL (nmol/L)	23.2 ± 14.8	25.9 ± 19.9	22.8 ± 12.7
Small VLDL (nmol/L)	30.2 ± 11.6	35.0 ± 14.5	34.7 ± 20.3
Total LDL (nmol/L)	1118.3 ± 263.8	1138.8 ± 255.9	1147.8 ± 226.8
IDL (nmol/L)	233.5 ± 126.3	213.4 ± 117.8	208.3 ± 115.8
Large LDL (nmol/L)	137.0 ± 116.6	164.0 ± 152.7	187.0 ± 142.0
Small LDL (nmol/L)	747.9 ± 209.4	755.6 ± 243.2	766.4 ± 203.7
Total HDL (μmol/L)	35.9 ± 5.5 ^a^	37.6 ± 7.1 ^b^	35.7 ± 6.0 ^a^
Large HDL (μmol/L)	6.1 ± 2.3 ^a^	6.6 ± 3.0 ^b^	5.9 ± 2.3 ^a^
Medium HDL (μmol/L)	11.0 ± 4.3	10.3 ± 4.4	9.7 ± 5.1
Small HDL (μmol/L)	18.8 ± 5.5	20.0 ± 5.2	20.1 ± 6.7
VLDL size (nm)	55.6 ± 7.4	54.5 ± 8.4	53.9 ± 10.4
LDL size (nm)	20.2 ± 0.5	20.3 ± 0.6	20.3 ± 0.5
HDL size (nm)	9.2 ± 0.4	9.2 ± 0.4	9.1 ± 0.4

* Values are presented as mean ± SD. Labeled without a common letter means differ at *p* < 0.05 by repeated-measures ANOVA with LSD post hoc analysis. (CM: chylomicrons; HDL: high-density lipoprotein; LDL: low-density lipoprotein; VLDL: very low-density lipoprotein).

**Table 6 nutrients-14-02138-t006:** Plasma choline and metabolites from fasting plasma of MetS participants (n = 24) at baseline and after EGG or SUB diet for 4 weeks each *.

Parameters (μmol/L)	Baseline	EGG	SUB
Choline	8.33 ± 2.08 ^a^	10.54 ± 2.8 ^b^	9.84 ± 3.17 ^a^
Betaine	35.94 ± 9.8 ^a^	43.4 ± 11.7 ^a^	39.1 ± 13.5 ^a^
DMG	2.25 ± 0.98 ^a^	3.06 ± 1.91 ^b^	2.8 ± 2.44 ^ab^
Methionine	30.6 ± 5.07	29.77 ± 8.3	30.4 ± 7.7
TMAO	2.3 ± 1.4 ^a^	2.8 ± 1.2 ^ab^	3.0 ± 2.05 ^b^

* Values are presented as mean ± SD. Values with different superscripts differ at *p* < 0.05 as determined by repeated-measures ANOVA. The statistics remain the same after adjusting for gender, age and Met-z score.

## Data Availability

Data for this study are available upon request to the Principal Investigator.

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
