# Peer review of "Eggs Improve Plasma Biomarkers in Patients with Metabolic Syndrome Following a Plant-Based Diet—A Randomized Crossover Study"

_nutrients, 2022, doi:10.3390/nu14102138_

Round 1

Reviewer 1 Report

Manuscript ID nutrients-1724473

Title: Eggs Improve Plasma Biomarkers in Patients with Metabolic Syndrome Following a Plant-Based Diet

The authors tested the hypothesis that the inclusion of whole eggs (EGG)  in combination with spinach in a plant-based diet would increase plasma concentrations of carotenoids and choline compared to an egg substitute and spinach. Additionally, we tested the hypothesis that the EGG intervention would increase plasma HDL and the formation of the larger HDL particles, allowing for the transport of more carotenoids in plasma to the various tissues. The ultimate hypothesis was that addition of whole eggs may represent a strategy to modulate biomarkers of disease.

General comment

The basic hypotheses are reasonable, however the quality of data supported to prove them is limited, questionable and scarce. In particular, the observed either limited or no changes in cholesterol concentrations in the lipoprotein subtractions, are hard to be accepted in the face of a 4-fold greater total dietary cholesterol intake. Low-density lipoproteins transport the majority of cholesterol in normal human plasma and distribute it to peripheral tissues, however this does not appear from the LDL-cholesterol data reported in the MS.

Major comments

  1. Following the above-reported general comment, the intake of plasma total cholesterol with the EGG diet is ≈4-fold (Table 3), whereas (Table 4) plasma HDL cholesterol, although reportedly statistically significant, is of very little magnitude (<3%), that of HDL <5%, while LDL-cholesterol is not increased at all. These data seem quite strange and odd. Perhaps the relatively short period of controlled diet administration wasn’t insufficient to produce relevant changes in plasma cholesterol sub fractions. The author should discuss and justify the emphasis attributed to the little changes of HDL cholesterol.
  2. Table 5: the item: “Large LDL” is reported twice…
  3. The baseline data apparently include those of both the EGG and the SUB group. They should rather be presented separately.
  4. Ln 60-61: “Daily whole egg consumption leads to more significant increases in plasma HDL-C and improvements in HDL profiles in MetS compared to intake of a yolk-free egg substitute during moderate carbohydrate restriction [9].” As a matter of fact, Ref. [9] does not report plasma concentration data.
  5. How do your results be reconciled with the overwhelming scientific literature suggesting a limitation of cholesterol consumption daily? (1 egg: =180 mg cholesterol…).

Minor comments

  1. Introduction: Please mention here what was the “egg substitute”.
  2. Ln 92: 0.3 mg of
  3. Table 5: “concentration”.
  4. There are several typographical errors.

Author Response

a point by point is provided in the attached file

Reviewer 2 Report

dear colleagues,

thank you for the paper. some suggestions to be addressed:

title and abstract – to be modified after main changes + in the title or abstract, utilize a generally recognized phrase to indicate the study's design.

keywords are repeat of tile use mesh to generate some better https://meshb-prev.nlm.nih.gov/

introduction – introduction need to be streamlined and reduce redundancy. explain the scientific context and justification for the reported research in mets.

methods – early in the text, present crucial features of research design. should be randomized crossover design. please follow consort http://www.consort-statement.org/

describe the context, places, and key dates, such as recruiting, exposure, follow-up, and data collection.

provide the qualifying criteria, as well as the sources and procedures of participant selection.

describe any steps you have made to address potential sources of bias e.g. race is not reported here and according to some criteria mets is ethnic- sex- group specific. according to ncep atp iii metabolic syndrome (based on 3 out of 5 characteristics: waist circumference. 88 cm for women and > 102 cm for men; plasma triglycerides > 150 mg/dl, blood pressure > 135/85 mm hg, fasting glucose > 100 mg/dl and hdl < 40 mg/dl for men and < 50 mg/dl for women). consider that you report it also according to idf if you want to be cited in future meta-analyses.

explain how the study size was arrived at. i suggest you discuss the results with a senior biostatistician. is this itt analysis? why parametric used when n = 24 only? can you pls report power analysis at alpha 0.05 and beta 0.2?

explain how quantitative variables were handled in the analyses.

describe all statistical techniques, especially those used to account for confounding variables.

describe any techniques used to investigate subgroups and interactions.

describe how missing data were handled.

mets z-score need to be reported at different times. this will show how severely mets they are.

describe any sensitivity analyses and if u can show roc/auc

give characteristics of study participants (eg demographic, clinical, social) and information on exposures and potential confounders. “21 caucasians and 3 african americans”

consider using infographic for the intervention if you want to brand it as thomas model

report other analyses done—eg analyses of subgroups and interactions, and sensitivity analyses

results – table 2 add sex + race (limitation smoking and sleep) any pts on meds? especially neuropsychiatric or steroidal tx? tally this with inclusions.

table 3 in methods its noted thar “dietary records were analyzed using the nutrition data system for research (ndsr) (nutrition coordinating center, university of minnesota), table 2 need better method description

table 4 – report mean arterial pressure + mets z score

in text i would like to see 95%ci

table 5 please recheck values they don’t add up also clarify that large ldl has two entries

fig 2 repeat data pls consider omit

table 6 please redo after controlling for age, sex, race and mets zscore at baseline using ancova

fig 3 low quality barely can be seen pleas enlarge

fig 4 clear but need in text ti point that r almost .6 so strong relationship

discussion – need to start with summary of results

report an important previous work by colleagues - thomas ms, dibella m, blesso cn, malysheva o, caudill m, sholola m, cooperstone jl, fernandez ml. comparison between egg intake versus choline supplementation on gut microbiota and plasma carotenoids in subjects with metabolic syndrome. nutrients. 2022 mar 11;14(6):1179. doi: 10.3390/nu14061179. pmid: 35334836; pmcid: pmc8951625.

also thomas ms, dibella m, blesso cn, malysheva o, caudill m, sholola m, cooperstone jl, fernandez ml. comparison between egg intake versus choline supplementation on gut microbiota and plasma carotenoids in subjects with metabolic syndrome. nutrients. 2022 mar 11;14(6):1179. doi: 10.3390/nu14061179. pmid: 35334836; pmcid: pmc8951625.

language need to be adjusted pls kindly tone down

i like to suggest two additional mini paragraphs

  • message for practice
  • your suggestions for future rcts

discussion can be shortened also by removing lit based knowledge

finally any muslims in ur study because i work on mets and ramadan fasting/tre/interm fasting as this/may have confounded ur results

Author Response

A point by point response is provided for reviewer 2

Round 2

Reviewer 2 Report

thanks for addressing my concerns